# Desmoid-Type Fibromatosis

**DOI:** 10.3390/cancers12071851

**Published:** 2020-07-09

**Authors:** Dorian Yarih Garcia-Ortega, Karla Susana Martín-Tellez, Mario Cuellar-Hubbe, Héctor Martínez-Said, Alethia Álvarez-Cano, Moises Brener-Chaoul, Jorge Adán Alegría-Baños, Jorge Luis Martínez-Tlahuel

**Affiliations:** 1National Cancer Institute (Instituto Nacional de Cancerologia), Mexico City 14080, Mexico; hubbemc@yahoo.com (M.C.-H.); mtzsaid@hotmail.com (H.M.-S.); 2Hospital Angeles de las Lomas, Huixquilucan, Estado de Mexico 52763, Mexico; karlamartin1@gmail.com (K.S.M.-T.); moises_bre@hotmail.com (M.B.-C.); 3Oncare Cancer Center, Monterrey Nuevo Leon 66220, Mexico; aalvarezuanl@gmail.com; 4National Cancer Institute (Instituto Nacional de Cancerologia), Mexico City 14080, Mexico; dr.alegria.unam@gmail.com

**Keywords:** desmoid tumor, desmoid-type fibromatosis, desmoid, β-catenin, aggressive fibromatosis

## Abstract

Desmoid tumors represent a rare entity of monoclonal origin characterized by locally aggressive behavior and inability to metastasize. Most cases present in a sporadic pattern and are characterized by a mutation in the CTNNB1 gene; while 5–15% show a hereditary pattern associated with APC gene mutation, both resulting in abnormal β-catenin accumulation within the cell. The most common sites of presentation are the extremities and the thoracic wall, whereas FAP associated cases present intra-abdominally or in the abdominal wall. Histopathological diagnosis is mandatory, and evaluation is guided with imaging studies ranging from ultrasound, computed tomography or magnetic resonance. Current approaches advocate for an initial active surveillance period due to the stabilization and even regression capacity of desmoid tumors. For progressive, symptomatic, or disabling cases, systemic treatment, radiotherapy or surgery may be used. This is a narrative review of this uncommon disease; we present current knowledge about molecular pathogenesis, diagnosis and treatment.

## 1. Introduction

Desmoid tumor (DT) refers to a rare benign neoplasm of monoclonal origin. Estimated incidence is 5–6 cases per 1,000,000 habitants per year and is more prevalent in 30–40 year-old patients. The clinical course is unpredictable, and local recurrences are common. [1,2]. The term desmoid was coined in 1838 by Mueller and arises from the Greek term *desmos*, hinting at its consistency and similarity to tendons. However, the first description of the disease was made in 1832 by MacFarlane. The World Health Organization (WHO) defines a DT as a clonal fibroblastic proliferation that arises in deep soft tissues and is characterized by infiltrative growth and a tendency towards local recurrence with no ability to metastasize [3]. DTs can occur anywhere in the body, but they are more frequent in the abdominal wall, intra-abdominal cavity, and limbs. Five to ten percent of cases are associated with familial adenomatous polyposis (FAP).

Some factors like trauma, surgery, pregnancy, and oral contraceptives have been associated with the appearance and progression of DTs, however the exact role of hormonal influence is not fully understood [4,5].

Even though about 90% percent of these tumors are sporadic, the remaining 5–10% present a hereditary pattern related to FAP. Patients with FAP have a 1000 times greater risk of developing DTs, and of the total number of patients with FAP, 5–16% will develop a DT [6,7].

Sporadic tumors develop most commonly in extra-abdominal locations, whereas those associated with FAP usually develop in the mesentery and/or in the abdominal wall. FAP-related DTs commonly face a more aggressive course and present with larger, multifocal tumors up to 10 years earlier than the sporadic variant [6,7,8].

## 2. Tumor Biology and Signaling Pathways Involved in the Oncogenesis of Desmoid Tumors

DT pathogenesis is strongly linked to the Wnt/β-catenin cascade, where β-catenin dysregulation plays a fundamental role. The Wnt/β-catenin pathway plays a role in the selection of cell destiny during fetal development and participates in adult homeostasis and regeneration processes. The genes targeted by this signaling pathway are involved in regulating the balance between self-renewal, differentiation, apoptosis, and cell maintenance. β-catenin is a molecule with multiple functions regulated by the APC gene and the Wnt pathway; Among them, it functions either as a cell adhesion molecule or as a nuclear transcription factor [4].

The APC gene plays a central role in the phosphorylation and proteosomal degradation of β-catenin; in turn, the Wnt pathway inhibits APC-dependent phosphorylation [9]. Depending on whether DT occurs sporadically or associated with FAP, there are 2 mutually exclusive mutations identified for the genesis of this pathology: the *CTNNB1* mutation in the former and the APC mutation in the latter (Figure 1) [10].

The *CTNNB1* gene is a proto-oncogene responsible for regulating cell adhesion and transcription. When mutated, abnormal stabilization and accumulation of β-catenin results, which in turn binds to the TBL1/TBLR1 protein, stimulating the expression of genes in the Wnt/APC/β-catenin pathway, including proliferation factors such as S100A4 or CTHRC1. Subsequently, the interaction between APC/β-catenin is modified inhibiting the ability of APC to degrade β-catenin. The β-catenin mutations are grouped in the N-terminal region in codons 32–45 encoded by exon, 3 being T41A and S45F, the most frequent found in 50 and 25% of cases respectively, and S45P the third, present in 9% of the cases [11,12].

In FAP associated DT, a non-functional APC gene is generated resulting in the excessive accumulation of intracellular β catenin. It stimulates genes such as cyclin-D1 or c-MYC that provoke cell proliferation and differentiation that culminate in the development of DT [13].

The better understanding of molecular genetics has contributed to an increased comprehension of the demeanor of this rare entity. Some studies found a correlation between S45F mutation and a higher risk of recurrences after surgery, as well as the increased possibility of progressive disease and poor response to meloxicam [14]. In the case of hereditary DT, the presence of the 5′ mutation in codon 400 is associated with a better prognosis compared to the 3′ mutation in codon 1400 in the APC gene [15].

## 3. Histopathology

Macroscopically DT are firm and gray or whitish, resembling scar tissue. On microscopy they are characterized by a heterogeneous, poorly defined and uniform proliferation of spindle cells that mirror myofibroblasts wrapped within a stroma of abundant collagen and a vascular network lacking capsule. Consistently, there is no atypia, necrosis or mitosis. The nuclei may contain euchromatin or heterochromatin. There is no histological difference found between FAP-related and sporadic DT. [16,17] In immunohistochemistry (IHC) they are characterized by nuclear positivity for β-catenin, vimentin, Cox2, c-KIT, PDGFRb, androgen receptors and beta estrogen receptors. They are negative for desmin, S-100, h-caldesmon, CD34 and CKIT [18,19].

## 4. Clinical Features

Presentation may vary from asymptomatic to disabling tumors. The variety of symptoms is in direct relation to the site, size, and progression speed. Tumors located in the limbs usually are symptomatic, especially when they grow nearby or infiltrate the neurovascular bundle, generating paresthesias, pain or polyneuropathies. Those in the intra-abdominal location grow asymptomatic until they reach large dimensions, causing intestinal obstruction, ischemia, and rarely even perforations or bleeding [20,21].

## 5. Image Studies

Radiological studies are essential not only for diagnosis but also for treatment and follow-up. Ultrasound, tomography and magnetic resonance are the main diagnostic tools. The appearance of tissue is variable depending on its composition which can be fibrotic, vascular or cellular according to the stage of the disease [22]. Ultrasound (US) serves as the initial evaluation especially for tumors in the extremities or abdominal wall. It is the study of choice for pregnant patients and to guide biopsies. The appearance differs from a smooth contour oval mass to a poorly defined soft tissue tumor with variable echogenicity [23]. There are two characteristic signs that can be identified: the “tail sign” is a linear extension into the fascial planes, and the “staghorn sign” caused by the intramuscular digitiform projections of the tumor [24].

In computed tomography (CT), appearance also varies from well-defined soft tissue mass (as in abdominal wall) to infiltrative margins (in mesenteric cases). They are isodense with the skeletal muscle with areas of hypo or hyper attenuation according to myxoid or fibrotic elements. Most of them bear a moderate contrast enhancement and do not display necrotic areas or calcifications. CT is considered the study of choice in the follow-up of patients with intra-abdominal desmoid tumors [25].

Magnetic resonance imaging (MRI) is the study of choice in extra-abdominal tumors and in patients allergic to iodine contrast agents. Its appearance depends on the cellular composition where more fibrotic areas, with a greater amount of collagen, show a low signal intensity in T2 with moderate contrast enhancement. Tumors with a greater amount of cellular or myxoid components are more heterogeneous with hyperintense areas in T2 sequences the “band sign” refers to the presence of linear bands within the tumor that do not enhance with contrast, hypointense in T1 and T2. It is found in 60–90% of the cases. Tumors with an intramuscular location are surrounded by a thin border of fatty tissue called the “split fat sign” or flame-shaped margins known as the “flame sign”. The behavior of DT on an MRI traduces its potential for progression. The association between tumor growth and T2 signal intensity has been described as being more biologically active in tumors when they show more than 90% hyperintense signal in T2, probably adding benefit from earlier interventions. When observed over time, tumors that decrease their intensity in T2 advise for a greater collagen deposition, therefore, it is a less active tumor. The same effect can be noticed in tumors that respond to treatment [26,27]. The utility of positron emission tomography (PET-CT) is not yet well established due to its low metabolic activity and inability to metastasize. Some reports advocate for its use in rapidly progressive cases to evaluate imatinib response [28].

## 6. Diagnosis

The diagnosis of DT begins with a high suspicion of the disease and early referral to a sarcoma center or a sarcoma surgeon is encouraged. Detailed physical examination, imaging studies, and biopsy should follow in accordance to the recommendations for soft tissue sarcomas. If surgical treatment is deemed necessary, it must include the biopsy site in the specimen, and be reviewed by an experienced pathologist, since up to 30–40% of cases are misdiagnosed [29].

## 7. Treatment

The current initial treatment strategy for DT advocates for an active surveillance period. Current recommendations are shown in the following algorithm (Figure 2) [29].

Active treatments in DT are multiple and include surgery, radiation therapy, systemic treatment or a combination of these. Comparisons have been made to better understand if any initial strategy is superior to others in terms of long-term disease control, results vary widely [29].

## 8. Active Surveillance

In recent years—as a result of the unsatisfactory results obtained with surgery—the initial approach has turned toward a conservative, non-operative strategy, which is now the strategy of choice. It is recommended that all cases have a period of active surveillance. It allows for better predictions concerning the natural history and biology of the disease and allows the clinician to plan the next step in the therapeutic sequence, since a high percentage of cases reach a stabilization period and some patients even present regression of the tumor [30]. Active surveillance was first proposed for patients that had recurrences not amenable for limb salvage; being a benign tumor, the main goal of observation was to avoid mutilating procedures. Under the surveillance period, it was noted that more than half of the tumors stabilized. Afterwards, the same strategy was offered to patients with resectable tumors and the same results were obtained, setting the foundations for what is now the initial treatment of choice; active surveillance [31].

The rationale for active surveillance is based on the fact that most patients will be able to avoid an unnecessary surgical procedure. Approximately 50% of the cases enter a stabilization period in an average of 14 to 19 months. When progressive disease presents, it is usually in the first months of observation and is rarely seen after three years of follow-up. Following this approach, only 14–16% of cases will require a surgical intervention and a quarter of the patients will show tumor regression. It has been demonstrated that the surveillance period can be safely carried out without detrimental outcomes in those patients who progress [32,33]. Recently, Duhil de Bénazé et al., reported encouraging outcomes in young patients with an initial wait and see approach. It was proved to be a safe and feasible option, not associated with impair in long term functionality when compared with other treatment strategies [34,35]. Another case-series, carried out by Fiore et al. included 142 patients (74 with primary tumors and 68 with recurrent tumors), in which a total of 83 patients were managed with active surveillance and reached a 5-year progression-free survival of 49.9% [35].

A study of patients with desmoid tumors managed at a referral center in the United Kingdom demonstrated a shift in the trend of treatment over time. The authors reported an increase from 10% in 1998 to 40% in 2016 of patients managed with active surveillance initially. In the whole series, they had a 36% of stable disease with 27% of either partial or complete response and 36% of progressive disease. They recorded older age (> 50 years old) as a risk factor for progression when compared with younger patients, as well as upper extremity and chest wall location [36].

The period of active surveillance should include close monitoring of patients with MRI or computed tomography every month for the first two months, then every three months for the first year followed by every six months until the fifth year, and yearly after. The intensity of the surveillance regimen, especially during the first years serves for early identification of rapidly progressive cases. Patients who have tumors in life-threatening locations as well as those with severe pain may avoid the surveillance period.

## 9. Surgical Treatment

Until a few years ago, surgical management was the standard of care for patients with DTs. The goal was to achieve negative margins; however, the 5-year recurrence rate was high, ranging from 25% to 60%, and extensive or disabling procedures were common. The role of the microscopically positive surgical margin is not consistently related to an increase in local recurrences (See Table 1: Influence of surgical margin on various series). In some studies, it has been proven an adverse factor, but it has not been reproducible in all the series, possibly due to the variability of the definition of surgical margin, the biology of the disease, and the retrospective nature of the studies. This is consistent in the series in which resections with negative margins are associated with recurrent disease, and patients with microscopically positive margins never relapse [36,37]. Amputations should seldom be used for DT, only in cases of unresectable and severe, untreatable symptomatic recurrent disease or when some treatment-related side effects (surgery +/− radiation therapy) cause significant loss of function or disabling chronic symptoms [38,39,40].

According to the latest European consensus on desmoid tumors, surgery should be considered in cases of progression to medical or radiation therapies, always considering location and age. When surgery is carried out, it should always be done trying to preserve function and after considering all the alternatives for it. Cases of mesenteric or retroperitoneal tumors not associated with familial polyposis can be treated initially with surgery due to the morbidity and symptoms they cause.

As for soft-tissue sarcomas, isolated limb perfusion is an option to be considered for those cases with progressive, unresectable disease or in which surgery would lead to significant function loss. It is especially useful in patients with distal or multifocal disease in whom systemic treatment has failed or who do not want/cannot receive it. The same agents, melphalan and tumor necrosis factor, are used and perfusion can be performed at different levels of the extremities. In one of the most relevant multicenter series, 25 patients were reported in whom 2 of them presented complete response, 16 partial responses and 7 stable disease with low toxicity [45]. Unlike other sarcomas, surgery for residual disease is not recommended for DTs.

## 10. Systemic Therapy

Current indications for systemic treatment include rapidly progressive disease or patient rejection to active surveillance. Systemic treatment options for DTs include non-steroid anti-inflammatory drugs (NSAIDs), anti-hormonal therapies, tyrosine kinase inhibitors. (TKI), and conventional “low dose” chemotherapeutic regimens, including liposomal doxorubicin.

Anti-inflammatories have shown the ability to block the β-catenin pathway mediated by COX2 or prostaglandins, and thus induce an objective response and improve pain control. The proposed mechanism of this response is due to COX2 overexpression in the tumor microenvironment, causing increased expression of platelet-derived growth factor (PDGF) that contributes to tumor growth, stimulates angiogenesis and promotes pathways of resistance to apoptosis. Among the most widely used anti-inflammatory drugs are sulindac, indomethacin, meloxicam and celecoxib [46].

Anti-hormonal agents have been used with favorable results even though we do not know precisely its mechanism of action. The pathway of transforming growth factor β may be implicated. Tamoxifen and toremifene are the most widely used agents. In one study, progression-free survival was 90% at 12 and 24 months. According to RECIST, partial response, stable disease, and disease progression were observed in 25%, 65% and 10% of patients, respectively. They can be used alone or in combination with anti-inflammatory (anti-COX2) drugs. They are generally the first line of treatment due to their good tolerance and low toxicity profile. Unfortunately, response is usually poor, mostly achieving stabilization of the disease and improvement in pain, which can be observed promptly after treatment instauration [47].

Controversy exists for the use of systemic chemotherapy in a disease that does not carry a metastatic potential; however, it should be considered as the first line treatment in patients with rapidly progressive or unresectable symptomatic tumors. Drugs used include doxorubicin (alone or in combination with dacarbazine), vinorelbine, vinblastine, and methotrexate. The anthracycline-based regimen is similar to the one used in sarcomas and is associated with high response rates. It is administered for 6–8 cycles, and regardless of the combination used, objective response or stabilization of the disease is achieved in 80% of the cases, with a lasting response in 45% of patients [48].

Tyrosine kinase inhibitors (TKI) have shown objective responses, despite the fact that their mechanism of action is not fully understood in this circumstance. Still, by blocking the receptor phosphorylation, activation, and proliferation of the kinase, they inhibit growth and block cell proliferation. Agents used include imatinib, nilotinib, sorafenib, sunitinib, and pazopanib. Imatinib, a selective TKI, inhibits several receptors including ABL, PDGFR, and CKIT and has demonstrated a 3-year progression-free survival of 58% with 6% regression after 19–26 months of treatment with a clinical benefit in 84% of the patients [49,50,51].

The DESMOPAZ trial deserves a special mention. It is a non-comparative, randomized, open phase 2 trial. Patients with progressive DT were included, and randomly assigned to pazopanib or vinblastine and methotrexate. The primary objective was the proportion of patients who did not progress in the first six months with 83.7% for pazopanib and 45% for methotrexate-vinblastine. Adverse events were well tolerated [52].

Gounder et al. conducted a phase 3, double-blind study in 87 patients with DT and progressive, symptomatic or recurrent disease. One arm received sorafenib 400 mg orally daily and was compared with placebo. Crossover to the sorafenib group was allowed for patients in the placebo group with progressive disease. The primary objective was progression-free survival. Objective response rates and adverse events were also evaluated. The 2-year progression-free survival rate was 81% in the sorafenib group and 36% in the placebo group. Before crossover, the objective response rate was 33% in the sorafenib group and 20% in the placebo group. Among the patients receiving sorafenib, the most frequently reported adverse events were grade 1 or 2 events, namely rash (73%), fatigue (67%), hypertension (55%), and diarrhea (51%). The high response rate in the placebo group is to be noted and can suggest a proportion of patients with spontaneous regression [53].

Among new therapies are those that involve the Notch pathway, which has a fundamental role in the differentiation of bone marrow, immunological and gastrointestinal cells and becomes relevant for its coordinated action with the Wnt route. Nonetheless, it has been reported that its effectiveness in DT is independent of the status of β-catenin. The mechanism of action of these drugs depends on the γ-secretase that binds to the intracellular portion of Notch and is translocated to the nucleus. It modulates and activates transcription factors. γ-secretase inhibitors block this pathway, thereby exerting its therapeutic effect on desmoid tumors [54].

Finally, several new pieces of evidence support the concept that deregulation of the mammalian target of the rapamycin (mTOR) cell proliferation/survival pathway may play an important role in tumor biology when the APC/β-catenin pathway is disrupted. Sirolimus, a drug that inhibits the mammalian target of rapamycin (mTOR), is currently being evaluated as an anti-cancer agent in desmoid tumor [54]. Table 2 summarizes the evidence for systemic treatments.

## 11. Radiotherapy

The role of radiotherapy is controversial. Indications for its use in DT are debated due to its toxicity, especially in the young population. In the Italian/French consensus it is recommended in progressive disease or in the absence of other therapeutic alternatives. [49] In a retrospective review of 22 articles the local control rate was 75% when radiotherapy + surgery was used, 78% for radiotherapy and 61% for surgery, including patients with positive and negative margins; therefore they suggest its use in anatomical sites where surgery can generate considerable morbidity, such as head and neck [64].

The recommended dose is 50–56 Gy in 2 Gy fractions. Other studies have published doses greater than 56 Gy, but they have failed to demonstrate improvement in local control and are associated with greater toxicity including edema, pathological fractures, fibrosis, soft tissue necrosis or vascular complications as well as radio-induced neoplasms [65]. In 2017 a meta-analysis was published and concluded that adjuvant radiotherapy should be considered especially in those patients with R1 or R2 resections, since they are at greatest risk of recurrence [66].

## 12. Follow up

It is recommended to monitor patients in an outpatient setting, to perform a physical examination and imaging every 3–4 months, for the first 2 years. Subsequently, the intervals may be longer, and the clinician may choose to alternate resonance and ultrasound. When active surveillance is chosen, it should be performed with magnetic resonance and with CT for intra-abdominal tumors every month the first 2 months, then every 3 months for a year followed by every 6 months for 5 years and annually thereafter.

## 13. Quality of Life

Patients with this disease are affected in all spheres (physical, social, cognitive and emotional). In addition, they are susceptible to periods of anxiety from diagnosis and during treatment due to uncertainty about therapeutic options [67,68]. Recently there has been a greater interest in the subject, and it is reflected in the creation of different organizations such as The Desmoid Tumor Research Foundation in the United States and the creation of patient advocacy groups in various institutions and social media [69,70]. Many of the series presented in this review revealed that pain might be a common indication for patients undergoing active surveillance to shift to a more aggressive treatment strategy namely surgery, systemic treatment or radiotherapy; therefore, it is strongly encouraged to keep in mind that since it is a benign tumor, it would be reasonable to measure a symptom-free survival period instead of progression free survival. It would only be achievable with a detailed evaluation of quality of life that should be done for every patient before, during and after any treatment. Patients with desmoid tumors have comparable levels of anxiety as sarcoma patients and as such they should be offered tools for distress and symptom screening and treatment.

There is no specific tool for evaluation of health-related quality of life in desmoid tumors patients. The DASH (disabilities of the arm, shoulder and hand) score, the Enneking/MSTS (Musculoskeletal Tumor Society) score, the TESS (Toronto Extremity Salvage Score) and the modified Johnstone scale are used only for extremity diseases, but not suitable for patients with tumors in other sites. Timbergen et al. identified key issues to address in six themes (diagnosis, treatment, follow up and recurrence, physical domain, psychological and emotional domain and social domain) that could form the basis of a future desmoid tumor specific tool; being the first three grouped as the process of healthcare and the other three as the symptoms and function [71]. Certainly, there is a deficit of resources directed towards this specific area that should be promptly addressed in order to have a true care for this group of patients; furthermore, as we better understand how patients cope with their disease, the better we will direct resources, therapies and improve functional and psychological outcomes [72].

## 14. Conclusions

DTs often cause significant functional limitations, pain, and even major disabilities. It affects young patients, so their quality of life during a productive and active age is severely affected, despite the fact that DTs do not cause distant metastasis. Early referral to a sarcoma center or sarcoma specialist should be encouraged. The unpredictable course, the low incidence of the disease and the lack of knowledge about it have been limiting factors toward a more rapid advancement of the management of this complex pathology. The integration of molecular biology with the clinical and therapeutic aspects will allow us to see each case as an individual entity, as part of a spectrum of manifestations of the disease and not as one heterogeneous group encumbered by such diverse results.

## Figures and Tables

**Figure 1 cancers-12-01851-f001:**
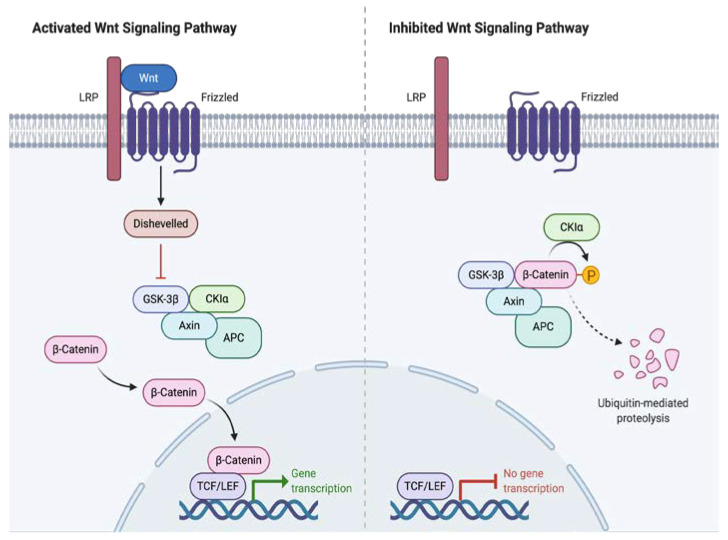
Wnt Signaling Pathway Activation and Inhibition.

**Figure 2 cancers-12-01851-f002:**
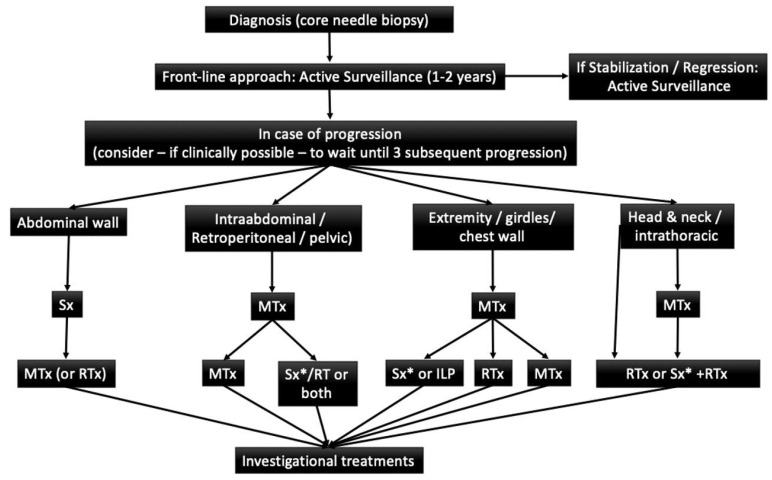
Adaptaded of European Consensus Initiative between Sarcoma PAtients EuroNet (SPAEN) and European Organization for Research and Treatment of Cancer (EORTC)/Soft Tissue and Bone Sarcoma Group (STBSG). (29) Sx: Surgery; MTx: Medical Treatment; RTx: Radiotherapy; ILP: Isolated limb perfusion * Surgery is an option if morbidity is limited.

**Table 1 cancers-12-01851-t001:** Influence of surgical margin on various series (5).

Author/Year	*N*	Primary/Recurrent	Median Months	5 Years DFS Margins (+)	5 Years DFS Margins (−)
**Posner, et al. 1989 [41]**	128	78/53	88	85	50
**Ballo, et al. 1999 [42]**	189	85/104	112	75	50
**Merchant, et al. 1999 [43]**	105	105/0	49	70	78
**Gronchi, et al. 2003 [39]**	203	128/75	130/153	82/65	79/47
**Huang, et al. 2009 [44]**	151	113/38	102	80	80
**Mullen, et al. 2012 [45]**	177	133/44	40	82	52
**Crago, et al. 2013 [46]**	57	382/113	60	69	69

* DFS. Disease-free survival.

**Table 2 cancers-12-01851-t002:** Evidence for systemic treatments in adults (39).

Treatment	Type of Study	*N*	Objective Response Rate	Other Response Rate	Reference
**Sulindac**	Retrospective	14	57%	-	[55]
**Toremifene**	Retrospective	27	22%	6-month PFS: 76%	[56]
**Metotrexate-Vinblastina**	phase II	27	15%	10-year PFS: 67%	[57]
**liposomal Doxorubicin**	Retrospective	14	33%	-	[58]
**Doxorubicin + dacarbazine**	Retrospective	12	50%	-	[59]
**Imatinib 800 mg/d**	phase II	51	6%	1-year PFS: 66%	[51]
**Imatinib 800 mg/d**	phase II	37	6%	6-month PFS: 65%	[60]
**Imatinib 400 mg/d**	phase II	50	12%	1-year PFS: 67%	[61]
**Sunitinib**	phase II	19	26%	1-year PFS: 80%	[62]
**Sorafenib**	Retrospective	26	26%	-	[63]

* PFS. Progression-free survival. Adapted Penel [39].

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
