# Peer review of "Desmoid-Type Fibromatosis"

_cancers, 2020, doi:10.3390/cancers12071851_

Round 1

Reviewer 1 Report

In this article, authors are reviewing desmoid type fibromatosis. Here they have compiled the molecular mechanism pathway involved, methods of diagnosis and treatment approaches. Here are some corrections/suggestions recommended to make it suitable for publication.

Abstract line 27 needs correction, also restructuring sentence would be helpful.

Signaling pathway representation with figure could be added.

Some abbreviations are needed full descriptions and Line 56 FAP already description given first time in the sentence line 56.

Author Response

.

We have received your comments with great enthusiasm.

  1. Line 27 has been changed. It referred to the non-familial presentation of the disease.
  2. Figure has been added.
  3. Abbreviations have been corrected.

Thank you for your time and kind contributions.

Reviewer 2 Report

The authors submit a narrative review on Desmoid type tumour. There is no clear aim of the paper whether it is a systematic review or just a narrative review. overall there is no clear, or observed new information coming out of the paper, to my knowledge. 

1. The authors needs to be clear on the purpose of the paper. is it a narrative review or a systematic review? there is no clear methodology of how the study is collated
2. There are no clear histological slides, there is need for this
3. Are authors able to report their local experience?
4. What are the factors affecting the long term prognosis, i.e. prognostic factors in such patients should be clearly determined
5. The conclusion is not support by the work, hence it needs revision

Author Response

We have received your comments with great enthusiasm, and we agree that further explanation should have been provided.

This paper was written at the magazine´s request, for a special number on rare tumors. The intention is to provide a narrative review of the available information regarding desmoid tumors.

1.It is a narrative review.

2.Authors work at a high-volume sarcoma center, however it was not the intention to include local experience, since the article is aimed to be a narrative review.

3.Regarding long term prognosis, there are few retrospective studies with more than 10yr follow up, so other than specific mutations (mentioned in the article), there is still not enough good quality data to answer that question, hopefully prospective analysis will answer it.

4.Conclusion has been corrected, however we do believe that statements made are common sense for physicians who treat and are familiarized with DT

We appreciate your comments and time dedicated to read this paper.

Reviewer 3 Report

This article is an extensive and complete review on diagnosis and management of desmoid tumors. I understand it should be included in a special issue entitled "New Therapeutic Advances in rare Tumors". To that extent, I wonder if the first parts like "Imaging studies" for example should be that detailed. The "active surveillance" part should probably be prioritized and not treated as the last paragraph of the treatment part since it is now the standard front-line approach. The biological and new treatment parts (TKI, notch inhibitor) should also be more detailed.

The relevance of working within network to imrpove the research and management of patients could be emphasized.  

Major comments:

The "systemic therapy" part on efficacy of active treatment should be interpreted according to data available on natural history of the disease (stable disease and objective response potentially related to spontaneous evolution more than treatment efficacy).

DT are not sarcoma and should not be considered as (line 238, 255, 261).

This reference should be listed: "The management of DT: a joint global consensus-based guidelines approach for adult and paediatric patients”, the Desmoid Tumor working group, Eur J Cancer 127 (2020) 96-107 (nice example of network publication).

Many mistakes in the references, not corresponding to the text or table.

Contradiction in line 251 with what is presented in Figure 1; even intra abdominal DT should benefit from active surveillance as front line approach

Minor comments:

line 27: "gene" instead of "gen"

line 60: remove the ";"

line 78-82: this part is confused: mutated beta-catenin is no longer degraded by the complex containing APC, leading to its accumulation in the cytoplasm and, then, translocation to the nucleus and gene expression stimulation.  

beta or b or ß-catenin but nor the 3 of them across the text.

Gene name should be italicized.

line 168: to my knowledge, no comparison of strategy has been published as first line treatment of DT. Ad references to that statement.

line 193: the unresectable characteristic of DT is not a criteria to start chemotherapy since surgical removal is no longer the standard treatment as explained above. 

line 200: which receptor are they talking about since these are multi targets TKI?

line 210: "adverse events were well tolerated" to correct

line 229-230: mTOR is blocked BY rapamycin, we don't block rapamycin

line 233: table 1 includes spanish words

Table 1: all references are wrong

Author Response

We have received your comments with great enthusiasm

1.- We believe that the “imaging studies” section is adequate, often this patient are first treated or evaluated by non-oncologist surgeons, and it is our hope that we can contribute to medical education, choosing a good imaging technique is important.

2.- “Active surveillance” was modified and placed as the first treatment strategy.

3.- New treatments have been detailed

4.- DT is not a sarcoma, this is clear. However, DT are classified as soft tissue tumors, and are treated by surgical oncologists, we intendend to present an analogy, not to stat that DT are sarcomas. Lines have been re-written to avoid this misunderstanding.

5.- Line 251 is about symptomatic intraabdominal tumors.

6.- Lines 78-82 have been rewritten

7.- References and writing were improved.

Round 2

Reviewer 2 Report

I think the authors tried to address some of my comments, yet it is a narrative review with limited addition to our knowledge.

Author Response

Dear reviewer:
As mentioned previously, this paper was requested by the journal for a special number on rare cancers. The objective is indeed to provide a narrative review of the current evidence available regarding desmoid tumors. Review articles are very useful specially for colleagues who do not deal with this diseases on a daily basis. Methodology did not include analysis on the data of the papers cited with the PRISMA guidelines for systematic reviews because is not a systematic review or meta-analysis. So, presenting the current evidence, as does any other narrative review published and not creating “new” knowledge is actually our objective.

The manuscript was reviewed by a company specialized in the translation of scientific texts.

Reviewer 3 Report

This draft of the article includes many of the suggestions proposed, the difeerent parts are better balanced, the messages more clear and the reading easier.

There are still several mistakes about references.

Please also take into acocunt the following comments:  

Line 97 : add « in the APC gene »

Line 105: which tyrosine kinase?

Lines 135-138: sentence difficult to understand

Line 188: reference #35 doesn’t correspond to the Fiore publication.

Lines 272-274: reference corresponding to imatinib assessment in DT is #51. You could also add #62

Lines 276-280: this part refers to #52 (several others references mistakes)

Line 309: “Penel et al.”

 Line 322: reference 67 correspond to an article published in 2000

Lines 330-332: already mentioned lines 198-200

Author Response

Dear Reviewer, I appreciate the suggestions:

Manuscript references were reviewed and corrected

The manuscript was reviewed by a company specialized in the translation of scientific texts.

Round 3

Reviewer 2 Report

The authors made the necessary changes